

# Species divergence in valleys: the phylogeny of *Phrynocephalus forsythii* complex and description of a new species

Qianru Liang and Lei Shi

Xinjiang Key Laboratory for Ecological Adaptation and Evolution of Extreme Environment Biology, College of Life Sciences, Xinjiang Agricultural University, Urumqi, Xinjiang, China

## ABSTRACT

**Background:** Geographic isolation caused by high-altitude valleys promotes the formation of geographic segregation of species, leading to species differentiation. The subgenus *Oreosaura* contains viviparous species from the Tibetan Plateau and the vicinity of the Tarim Basin, which can be divided into three species complexes according to their geographical distribution: *Phrynocephalus vlangalii*, *Phrynocephalus theobaldi*, and *Phrynocephalus forsythii*. However, molecular data for the *P. forsythii* complex are limited and the diversity of this species complex has been greatly underestimated. Therefore, this study aimed to explore the species diversity of *Oreosaura* and species differentiation within the *P. forsythii* complex.

**Methods:** We analysed the species diversity of *Oreosaura* by combining previous data, constructed a phylogenetic tree of the subgenus based on cytochrome c oxidase subunit I and 16S sequences, and estimated the divergence time.

**Results:** The results suggest significant genetic differences between the Tarim Basin populations and adjacent mountain valley populations of the *P. forsythii* complex and that the combination of deep valley landscapes in the high mountains and ice-age events have contributed to the differentiation of the viviparous toad-headed agama lizard, which is a key factor in the phylogenetics of the *P. forsythii* complex. Furthermore, we identified a population collected from Wuqia County, Xinjiang, as a new species, *Phrynocephalus kangsuensis* **sp. nov.** The results will provide data for phylogenetic studies following the *P. forsythii* complex and help demonstrate that valleys promote the formation of *Phrynocephalus* species.

## INTRODUCTION

Geographic isolation results in an inability to exchange genes between populations and can eventually lead to population differentiation over time and high selection pressure (*Coyne & Orr, 2004*; *Nosil, Harmon & Seehausen, 2009*). Thus, pronounced geographical isolation promotes species differentiation (*Mayr, 1970*; *Kadmon & Pulliam, 1993*; *Coyne & Orr, 2004*; *Nosil, Harmon & Seehausen, 2009*; *Winger & Bates, 2015*). In addition to island habitats, high mountains and valley topography on continents geographically isolate

Corresponding author
Lei Shi, leis@xjau.edu.cn

populations, promoting species differentiation (*Wang et al., 2021*, *2022*; *Bogdanov et al., 2023*; *Nazarov et al., 2023*).

The subgenus of *Phrynocephalus, Oreosaura*, was established by *Barabanov & Ananjeva (2007)* and contains the species complex of *Phrynocephalus vlangalii*, *Phrynocephalus theobaldi*, and *Phrynocephalus forsythii*. This subgenus contains viviparous species that live at high altitudes on the Tibetan Plateau and low altitudes in the Tarim Basin. Among them, only *P. forsythii* has both valleys populations living in the high mountains and valleys and plains populations distributed in the basins, rendering it ideal for studying species differentiation.

Early phylogenetic studies of *Phrynocephalus* found that *P. forsythii* was the first to diverge in the viviparous species complex, forming a sister clade with another viviparous species complex (*Pang et al., 2003*; *Guo & Wang, 2007*; *Jin & Brown, 2013*). *Qi et al. (2020)* conducted genetic analyses of *P. forsythii* populations distributed in the Tarim Basin and found that the geographic populations form sister clades and have a monophyletic lineage, suggesting that *P. forsythii* populations distributed around the Tarim Basin may be a circumscribed species, with the results showing no obvious genetic differentiation among these populations distributed at low altitudes; however, their study did not obtain data on populations of *P. forsythii* at high altitudes. In recent years, we collected several valleys *P. forsythii* populations on field trips and found significant morphological differences from plains populations. *Phrynocephalus nasatus* is closely related to *P. forsythii* and its dorsal body colour pattern is similar to that of *P. forsythii* (*Golubev & Dunayev, 1995*; *Zhao, Zhao & Zhou, 1999*). *Phrynocephalus nasatus* has not been reported since its discovery and description as a new species by the Zoological Museum of Moscow University in 1995 (*Golubev & Dunayev, 1995*) and there have been no further studies on the holotype despite its distinctive morphological characteristics and questionable taxonomic status. *Phrynocephalus nasatus* has been described as a synonym of *P. axillaris* (*Barabanov & Ananjeva, 2007*). *Dunayev (2020)* described the habitat, ecology, and morphological body colour of *P. nasatus* but did not make morphological comparisons with *P. forsythii*. In their analysis of *Oreosaura*, *Solovyeva et al. (2023)* found that *P. nasatus* was closely related to *P. forsythii* in phylogenetic relationships.

*Solovyeva et al. (2018)* combined mtDNA and nuDNA to analyse the impact of environmental change on the evolutionary and biogeographical history of *Phrynocephalus* and concluded that the divergence of *Oreosaura* was related to the timing of the major uplift of the Tibetan Plateau. Recently, *Solovyeva et al. (2023)* constructed a very detailed phylogenetic tree for *Phrynocephalus* based on cytochrome oxidase subunit I (COI) sequences, in which *Oreosaura* was described in detail; however, because of the limited molecular data for the *P. forsythii* complex, the genetic differentiation of this complex was not discussed in their study.

Thus, the present study aimed to explore the species diversity of *Oreosaura* and species differentiation within the *P. forsythii* complex based on the data of *Solovyeva et al. (2023)* and with additional data from valleys areas of *P. forsythii* and 16S sequences. We explored the relationship between the phylogenetic lineages of the *P. forsythii* complex and the main

tectonically induced events in southern Xinjiang. We believe that the phylogenetic relationships of the *P. forsythii* complex can be used to evaluate the influence of paleogeographic and climatic factors.

## MATERIALS AND METHODS

### Materials

We collected specimens from three geographically segregated populations: altitude > 2,000 m (higher altitude populations) distributed in the valley. All the specimens (18 in total) we used were fixed in 95% ethanol, and were collected from three high-altitude areas (altitude > 2,000) (Table S1)—the Kangsu River (Wuqia population, altitude: 2,717 m) and Toshkan River (Akqi population, Group AKQ, altitude: 2,125 m) valleys in southern Tien Shan, and the Karakash River (Hotan population, Group HT, altitude: 3,148 m) valley in the middle Kunlun Mountains in 2006 and 2009 (Fig. 1).

The liver or muscle tissue in the samples was preserved with 95% ethanol for DNA extraction. All specimens were deposited in the Animal Herbarium of the College of Life Sciences, Xinjiang Agricultural University. These experimental procedures involving animals were approved (animal protocol number: 2023013) by the Animal Welfare and Ethics Committee of Xinjiang Agricultural University, Urumqi, Xinjiang, China. We also adhered to the ARROW (Animals in Research: Reporting On Wildlife) guidelines.

Sequences available for other species of the subgenus *Oreosaura* were collected and downloaded from the GenBank at the National Center for Biotechnology Information and Barcode of Life Database (Table S1).

### DNA extraction and polymerase chain reaction (PCR) amplification

Total DNA was extracted from tissue samples using the Foregene Animal Tissue Genomic DNA Extraction Kit, according to the manufacturer's instructions. The mitochondrial COI and 16S rRNA genes (16S) were used as DNA barcode markers and amplified by Polymerase Chain Reaction (PCR) using published primers and cycling parameters (*Pavlicev & Mayer, 2009*; *Nagy et al., 2012*). The PCR reaction system volume was 25 μl, which contained 1 μl of template DNA, 1 μl each of upstream and downstream primers, 12.5 μl of 2x Taq PCR Mix, and 9.5 μl of ddH$_2$O. The PCR conditions for the COI gene were as follows: 94 °C, 4 min; 94 °C, 30 s; 52 °C, 30 s; 72 °C, 50 s; 35 cycles; and 72 °C, 10 min. The PCR conditions for the 16S gene were as follows: 94 °C, 2 min; 95 °C, 30 s; 59 °C, 30 s; 72 °C, 50 s; 35 cycles; and 72 °C, 7 min. Both amplicons were stored at 4 °C after the PCR was completed. Subsequently, gel electrophoresis with 0.5% TBE (Tris-Borate-EDTA buffer) solution and agarose was performed to confirm successful amplification. Successfully amplified PCR stock solution was sent to Sangon Biotechnology (Shanghai, China) for purification and sequencing.

### Phylogenetic analyses & divergence time estimation

The sequencing results were comparatively spliced and manually edited using SeqMan with DNASTAR v6 (*Burland, 1999*). The spliced sequences were aligned using MEGA 7.0.26 (*Kumar, Stecher & Tamura, 2016*) and were converted into amino acid sequences

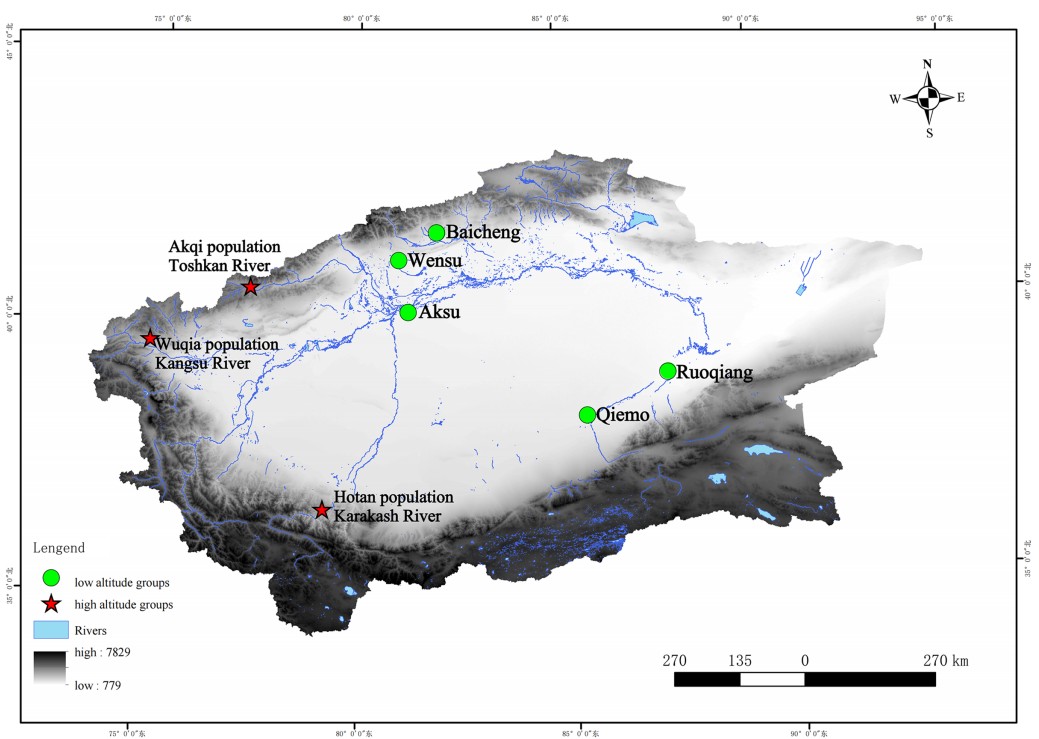

**Figure 1 Geographic characteristics and locations of the sampling sites and reference sequences of this study.** Note: For names of the areas concerned, please refer to Table S1. Stars, sample sites collected in this study (high-altitude); circles, sample sites of some of the reference sequences (low-altitude).                                   

using protein-coding to check for stop codons. Subsequently, genetic distances (*p*-distances) were calculated using MEGA 7.0.26. We used Partitionfinder-2.1.1 (*Lanfear et al., 2017*) to estimate the best model for the molecular dataset using the Akaike information criterion. The best models for COI sequences were the HKY, GTR+G, and SYM+I+G models, the best model for 16S sequences was the TIM+I+G model, and the best models for spliced sequences were the TRN+I+G, HKY, and GTR models. We then used MrBayes 3.2.2 (*Ronquist et al., 2012*) and PhyloSuite 1.2.2 (*Zhang et al., 2020*) to perform Bayesian inference (BI) and maximum likelihood (ML) on the three datasets. Phylogenetic trees were constructed using *P. mystaceus* and *P. axillaris* as outgroups. We used the Markov chain Monte Carlo (MCMC) theory approach in BI, running 5,000,000 generations, drawing one sample every 1,000, running four chains per analysis, and removing the top 25% of the data with the other parameters as the default. We used the K80 (K2P) method in ML, running 1,000,000 generations with the other parameters set as their default values (*Zhang et al., 2020*). The final results were checked using Tracer 1.7.2 (*Rambaut et al., 2018*) and had effective sample size (ESS) values > 200 for all parameter values. Finally, a phylogenetic tree was plotted using ITOLV6 (*Letunic & Bork, 2021*). We considered an *a posteriori* probability (BPP) value of > 0.99 or maximum likelihood bootstrap value of > 95% as strong support for monophyly.

We used the mtDNA dataset (COI+16S) to estimate species divergence times within the subgenus *Oreosaura* using BEAST 2.6.6 (*Bouckaert et al., 2014*) based on the lognormal

relaxed clock assumption. We used two calibration points: (1) the Pamir-Tian Shan uplift 10 million years ago (Ma) (*Charreau et al., 2009*) and (2) the final uplift of the Tibetan Plateau 3.6 Ma (*Cui et al., 1996*). We used the GTR and Yule models and sampled every 10,000 iterations, with 25% of the initial samples discarded as aged samples. Tracer 1.7.2 was used to evaluate the estimated ESS for all the parameters.

## Species delimitation

We followed the phylogenetic species concept and used COI sequences to classify molecular operational taxonomic units (MOTUs) into putative species. To assess the number of putative species-level lineages in the subgenus *Oreosaura*, we referred to the results of *Solovyeva et al. (2023)* and used the following three automated phylogenetic species concept-based species delimitation methods to estimate the number of species in the barcode data: (1) the automatic barcode gap discovery (ABGD) (*Puillandre et al., 2012*), (2) assemble species by automatic partitioning (ASAP) (*Puillandre, Brouillet & Achaz, 2021*), and (3) a Bayesian implementation of Poisson tree processes (bPTP) (*Zhang et al., 2013*).

In ABGD (https://bioinfo.mnhn.fr/abi/public/abgd/abgdweb.html) and ASAP (https://bioinfo.mnhn.fr/abi/public/asap/asapweb.html), the Kimura (K80) TS/TV (2.0) model was used, with other parameters defaulted in ABGD, 20 selected in steps and one in X (relative gap width) in ASAP, and the remaining default parameters. In bPTP (https://species.h-its.org/ptp/), the thinning selected 200 and the remaining parameters were set to default.

## Morphometry & scale statistics

Morphometric measurements were performed on *P. forsythii* specimens (six in total) collected from geographic populations (Wuqia) at valleys. Seven morphometric indices were measured using 0.01 mm vernier callipers, including snout-vent length (SVL), head length (HL), maximum head width (MHW), head height (HH), rostral eye distance (RED), nasal distance (NBD), and tail length (TL). The nasal scales (NS), inter-nasal scales (IS), supralabial shields, and infralabial shields were also counted. Morphological descriptions of *P. forsythii* and *P. nasatus* from the literature (*Zhao, Zhao & Zhou, 1999*; *Dunayev, 2020*) were also collected for comparison with the samples from this study.

## Zoobank registration

The electronic version of this article in the Portable Document Format (PDF) represents a published work according to the International Commission on Zoological Nomenclature (ICZN); hence, the new names contained in the electronic version are effectively published under that code from the electronic edition alone. This published work and the nomenclature it contains have been registered in ZooBank, an online registration system for ICZN. The ZooBank Life Science Identifiers (LSIDs) can be resolved and the associated information can be viewed through any standard web browser by appending the LSID to the prefix http://zoobank.org/. Publication LSID urn:lsid:zoobank.org:pub:812FA766-6BC6-4D39-85B9-8E2C58E06098 *Phrynocephalus kangsuensis* sp. nov.: urn:lsid:zoobank.org:act:53EF51A0-DEEA-4E6A-B62F-AD61AB3868E9. The online version of this work

# RESULTS

## Sequence & phylogenetic tree

The base sequences of two mitochondrial target fragments, 16S (475 bp) and COI sequences (623 bp), and a spliced dataset (1,099 bp) were obtained by extracting and sequencing DNA from three valleys geographic populations of *P. forsythii*. The results of the COI phylogenetic tree (Fig. 2) showed that the viviparous species complex (*Oreosaura*) was monophyletic (BPP/BS: 1.00/100). The *P. forsythii* complex first differentiated in *Oreosaura* and was monophyletic (BPP/BS: 1.00/96), with the geographic populations of the three valleys divided into three subclades (lineages A, B, and C). The Akqi population (Lineage C) formed a sister clade to the lower altitude populations (Lineage D, containing *P. nasatus*), and the Wuqia population (Lineage A) formed a sister clade to the other clades. *Phrynocephalus nasatus* is present in the low-altitude *P. forsythii* (Lineage D). The 16S phylogenetic tree (Fig. 3) shows the same topological complementary structure as the COI phylogenetic tree: *Oreosaura* and the *P. forsythii* complex are monophyletic (BPP/BS: 1.00/100; BPP/BS: 1.00/99, respectively). The Wuqia population (Lineage A) was monophyletic (BPP/BS: 1.00/100) and formed a sister clade with the Hotan population (Lineage B). Similarly, with the same structure as the splice sequence (Fig. 4), the Wuqia population (branch A) was monophyletic in the *P. forsythii* complex (BPP/BS: 1.00/100).

## Genetic distance

Uncorrected *p*-distance based on COI sequences (Table 1) showed that species in the subgenus *Oreosaura* were genetically distant from species in other subgenera (*P. axillaris* and *P. mystaceus*) by 12.5–18.5%, and between species within subgenera by 0.0–11.1%. In the *P. forsythii* complex, there were significant differences in genetic distances (1.4–5.2%) between the valleys and plains geographic populations. The population from Wuqia (*P. kangsuensis* **sp.nov.**) was genetically distant from geographic populations of the other two valleys (Group HT and Group AKQ) by 5.5–5.8%, and up to 5.2% from *P. forsythi*. No difference was noted in the genetic distance between *P. nasatus* and *P. forsythii* (0.0%). In addition, uncorrected *p*-distance based on 16S sequences showed significant differences in genetic distances between species within the subgenus *Oreosaura* (0.0–4.9%). In the *P. forsythii* complex, genetic differences between the valleys and plains populations ranged from 2.2–0.5%, with the Wuqia populations (*P. kangsuensis* **sp.nov.**) being significantly genetically different from the other populations (2.0–2.3%).

## Species delimitation

Species delimitation of the subgenus *Oreosaura* was defined based on COI sequences using three species delimitation methods (ABGD, ASAP, and bPTP) (Fig. 2). All three methods defined the Wuqia population (Lineage A) and Hotan population (Lineage B) as MOTUs. ASAP considered 11 species to be present in the subgenus *Oreosaura*, with the Akqi population (Lineage C), low-altitude population (Lineage D), Wuqia population (Lineage

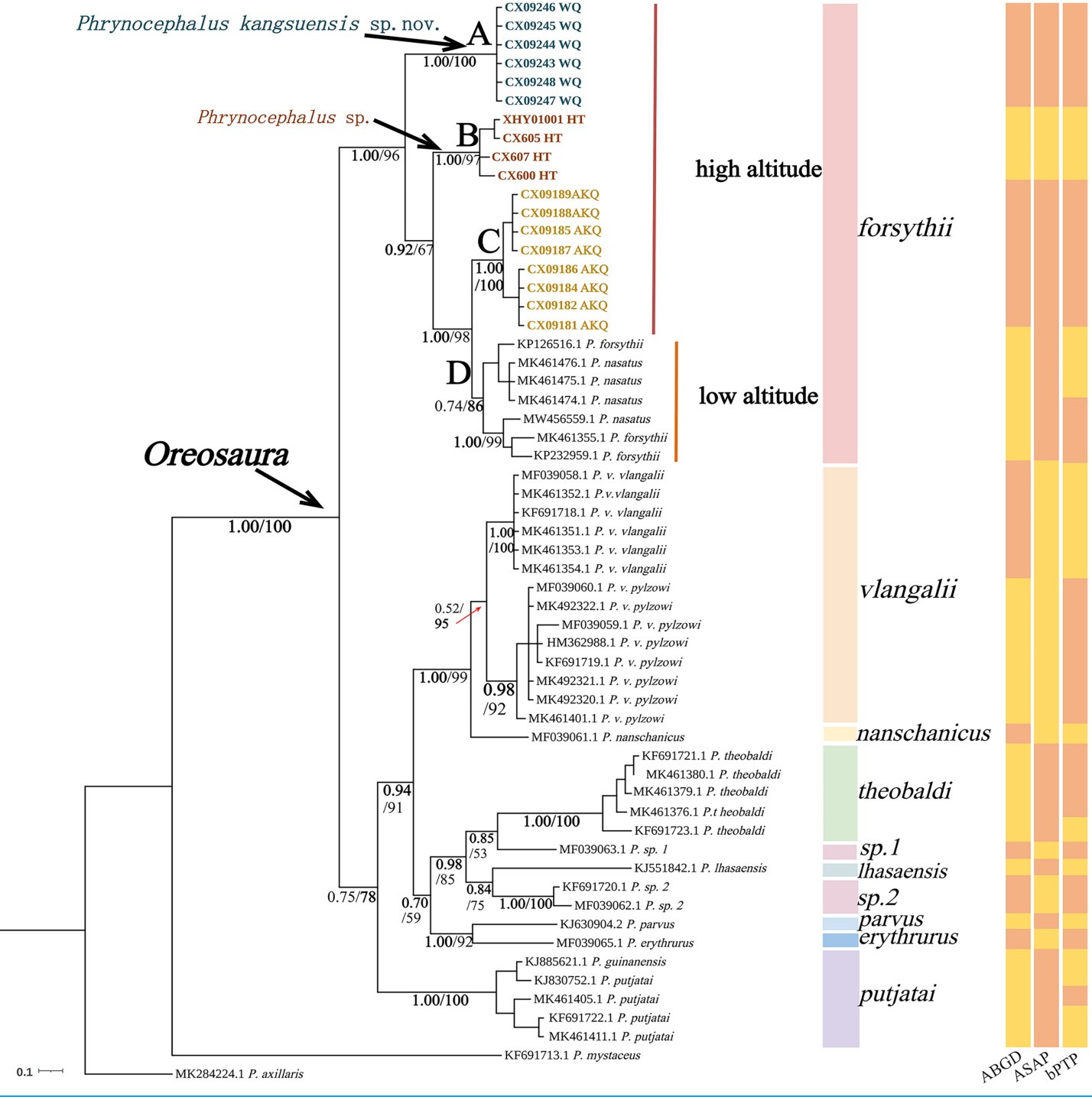

**Figure 2 Phylogenetic tree of COI sequence of *Oreosaura*.** The value on the node is BPP/BS, coloured sections are the species divisions of *Solovyeva et al. (2023)*. The highest support values are in bold.

A), and Hotan population (Lineage B) in the species group of *P. forsythii* considered to be three MOTUs and *P. vlangalii* and *P. nanschanicus* considered to be a MOTU. In contrast, bPTP determined 18 MOTUs.

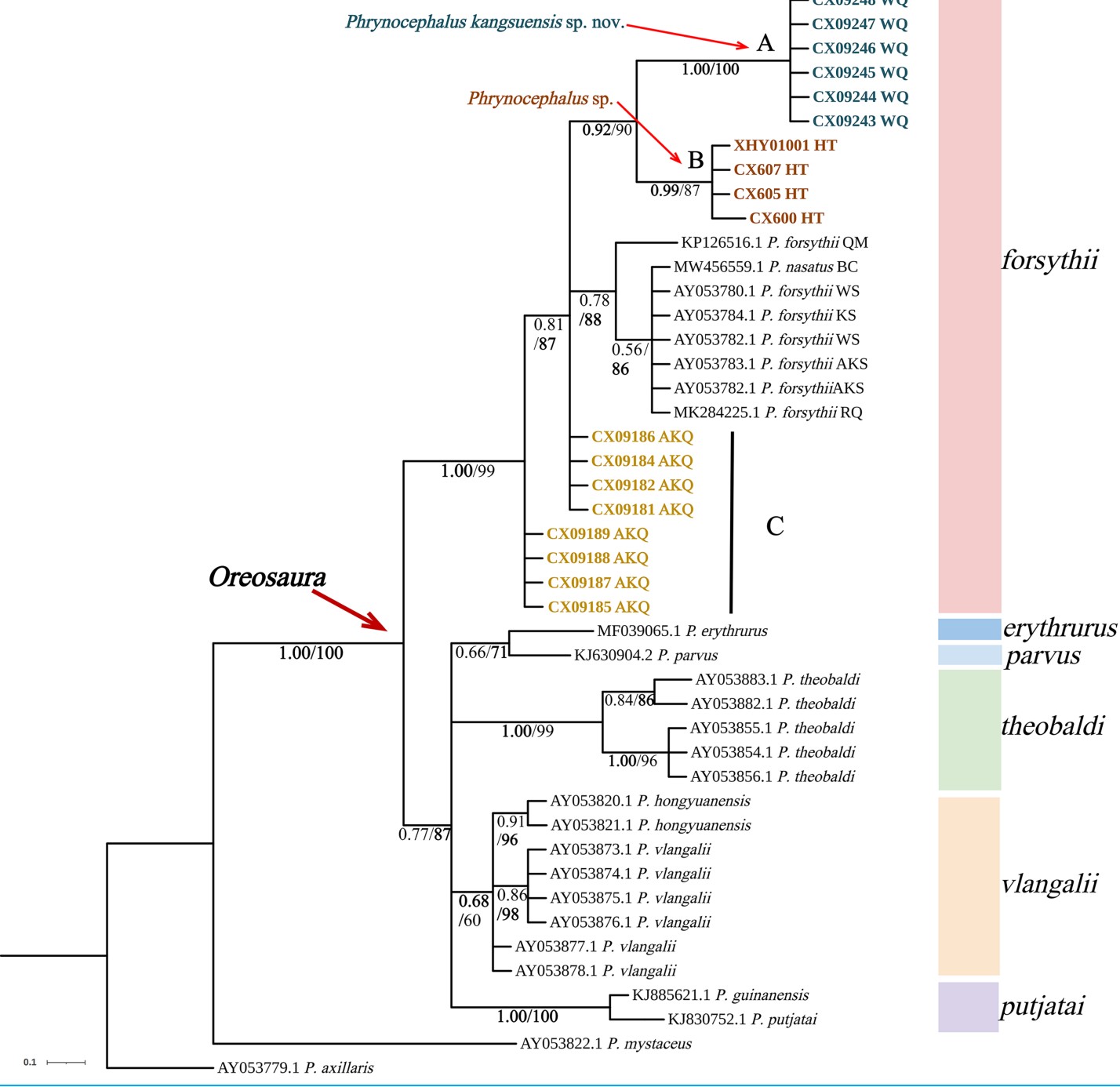

**Figure 3 Phylogenetic tree of 16S sequence of *Oreosaura*.** The value on the node is BPP/BS. Note: QM, Qiemo; BC, Baicheng; WS, Wensu; KS, Kashgar; AKS, Aksu; RQ, Ruoqiang. The highest support values is in bold.

## Divergence time estimates

The divergence times for subgenus *Oreosaura* and geographic populations of *P. forsythii* based on spliced sequences were estimated (Fig. 5). *Oreosaura* diverged at 10.43 Ma (95% HPD: [10.00–11.33] Ma). The *P. forsythii* complex diverged at 6.73 Ma (95% HPD:

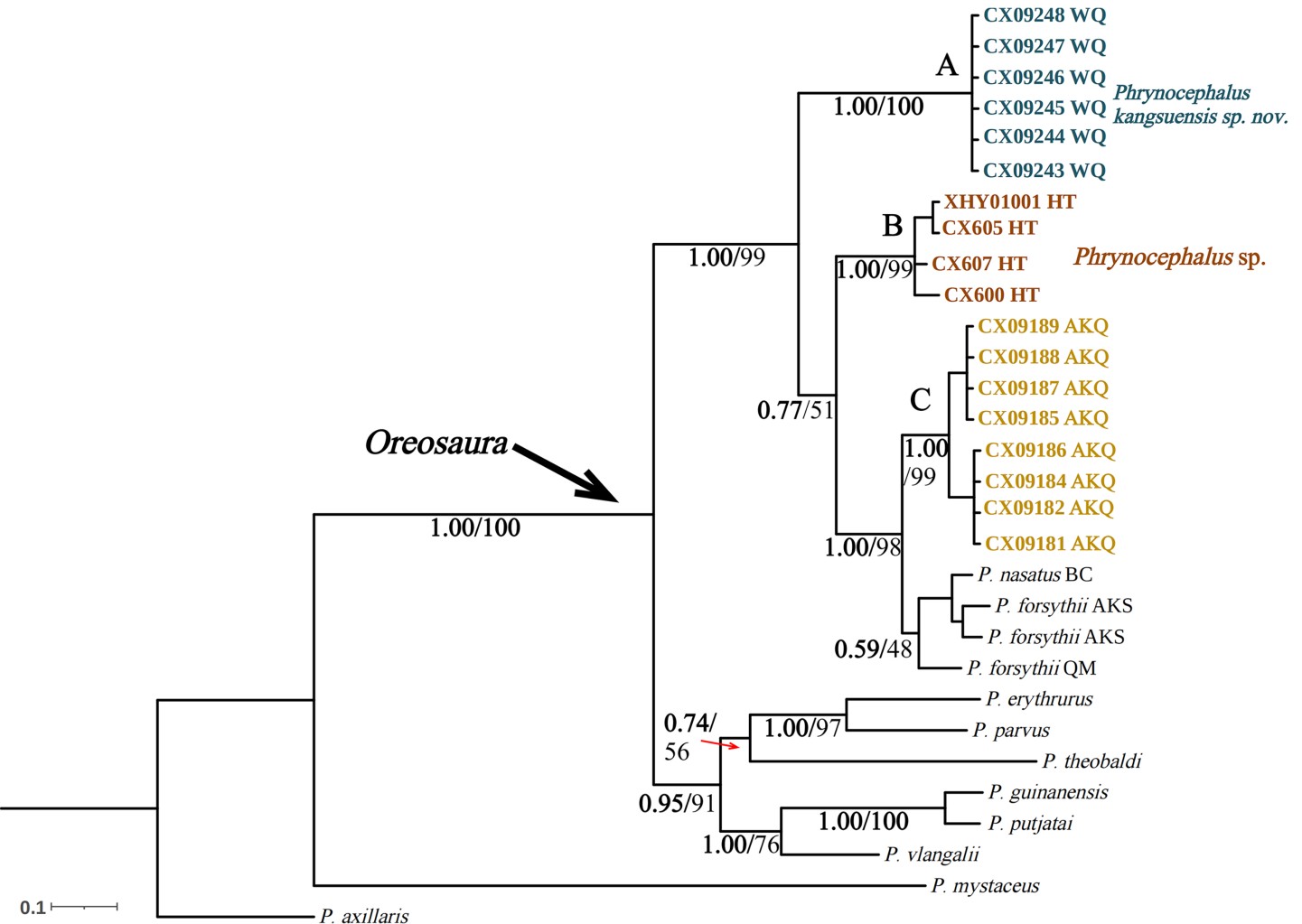

**Figure 4  Bayesian phylogenetic trees of *Oreosaura* on spliced sequences (CO1 and 16S).** The value on the node is BPP/BS. Note: QM, Qiemo; BC, Baicheng; AKS, Aksu. The highest support values are shown in bold.     

[4.75–8.76] Ma). The Wuqia population (*P. kangsuensis* **sp.nov.**) was the first within the complex *P. forsythii*, at 3.96 Ma (95% HPD: [2.57–5.40] Ma), followed by the Hotan population at 2.92 Ma (95% HPD: [1.75–4.16] Ma). The divergence between low- and valleys populations (Akqi population) occurred at 1.59 Ma (95% HPD: [0.88–2.35] Ma).

## Description of the new species
### *Phrynocephalus kangsuensis* sp. nov.

**ZooBank:** lsid:zoobank.org:act:812FA766-6BC6-4D39-85B9-8E2C58E06098.

**Specimens:** The holotype was male (field collection no. CX09247) and was collected by Lei Shi from Kangsu Town, Wuqia County, Kizilsu-Kirghiz Autonomous Prefecture, Xinjiang. The allotype was female (CX09246) and the paratypes consisted of four males (CX09243, CX09244, CX09245, and CX09248), all of which were collected from the vicinity of Kangsu Town, Wuqia County, Xinjiang, which has an altitude of 2,300 m above sea level.

**Table 1 Uncorrected *p*-distances (%) between species of *Oreosaura*.**

| | 1 | 2 | 3 | 4 | 5 | 6 | 7 | 8 | 9 | 10 | 11 | 12 | 13 | 14 | 15 | 16 | 17 | 18 |
|---|---|---|---|---|---|---|---|---|---|---|---|---|---|---|---|---|---|---|
| 1. *P. kangsuensis* **sp.nov.** | | 2.0 | 2.1 | 2.2 | 2.3 | – | 3.3 | – | 4.6 | – | – | – | 4.1 | 4.9 | 4.3 | 4.1 | 5.9 | 7.6 |
| 2. Group AKQ | 5.5 | | 1.4 | 1.7 | 1.8 | – | 2.3 | – | 3.5 | – | – | – | 3.0 | 3.8 | 3.3 | 3.0 | 5.7 | 6.5 |
| 3. Group HT | 5.8 | 3.5 | | 0.5 | 0.6 | – | 2.0 | – | 2.3 | – | – | – | 2.6 | 2.9 | 3.4 | 3.1 | 5.0 | 6.0 |
| 4. *P. forsythii* | 5.2 | 1.4 | 2.6 | | 0.0 | – | 2.5 | – | 2.9 | – | – | – | 3.2 | 3.5 | 3.5 | 3.2 | 4.6 | 6.7 |
| 5. *P. nasatus* | 5.4 | 1.6 | 3.0 | 0.0 | | – | 2.6 | – | 3.0 | – | – | – | 3.3 | 3.5 | 3.5 | 3.3 | 4.6 | 6.7 |
| 6. *P. v. pylzowi* | 8.5 | 7.2 | 8.5 | 6.9 | 7.0 | | – | – | – | – | – | – | – | – | – | – | – | – |
| 7. *P. vlangalii* | 7.7 | 7.1 | 7.8 | 6.6 | 7.0 | 1.2 | | – | 1.5 | – | – | – | 0.9 | 1.6 | 1.3 | 1.1 | 4.8 | 5.3 |
| 8. *P. nanschanicus* | 8.1 | 7.1 | 8.3 | 7.1 | 7.5 | 2.4 | 2.4 | | – | – | – | – | – | – | – | – | – | – |
| 9. *P. theobaldi* | 9.6 | 10.0 | 10.0 | 8.9 | 9.6 | 5.9 | 6.5 | 7.0 | | – | – | – | 2.0 | 2.0 | 3.1 | 2.8 | 5.3 | 5.3 |
| 10. *P. sp. 1* | 8.0 | 7.4 | 7.4 | 6.5 | 6.6 | 5.9 | 5.2 | 5.8 | 5.9 | | – | – | – | – | – | – | – | – |
| 11. *P. lhasaensis* | 11.1 | 10.1 | 10.8 | 8.1 | 8.5 | 7.7 | 7.8 | 7.1 | 7.0 | 6.1 | | – | – | – | – | – | – | – |
| 12. *P. sp. 2* | 9.1 | 7.8 | 7.8 | 6.8 | 7.4 | 6.7 | 6.1 | 6.1 | 6.4 | 4.5 | 4.8 | | – | – | – | – | – | – |
| 13. *P. parvus* | 8.7 | 9.4 | 9.4 | 8.5 | 9.0 | 6.1 | 5.5 | 6.1 | 6.9 | 5.8 | 8.1 | 6.1 | | 1.2 | 2.3 | 2.5 | 5.4 | 5.4 |
| 14. *P. erythrurus* | 8.7 | 8.0 | 8.1 | 7.1 | 7.7 | 6.7 | 6.8 | 6.1 | 8.3 | 5.8 | 7.4 | 5.5 | 4.8 | | 2.8 | 3.0 | 5.7 | 5.6 |
| 15. *P. putjatai* | 7.8 | 7.1 | 7.3 | 6.5 | 6.5 | 5.9 | 6.3 | 7.0 | 6.1 | 5.7 | 7.7 | 7.1 | 8.3 | 9.2 | | 0.2 | 6.2 | 5.9 |
| 16. *P. guinanensis* | 7.7 | 6.8 | 7.1 | 6.0 | 5.8 | 7.2 | 7.1 | 7.9 | 7.4 | 6.1 | 8.1 | 7.1 | 9.1 | 9.4 | 0.7 | | 5.9 | 6.2 |
| 17. *P. axillaris* | 13.9 | 12.8 | 13.0 | 12.5 | 13.0 | 13.9 | 13.9 | 12.5 | 17.1 | 13.9 | 15.4 | 14.3 | 13.5 | 13.5 | 13.6 | 13.2 | | 5.4 |
| 18. *P. mystaceus* | 15.0 | 13.4 | 14.7 | 13.3 | 14.0 | 16.2 | 15.9 | 15.8 | 18.5 | 14.7 | 15.4 | 15.8 | 14.6 | 14.2 | 15.0 | 14.7 | 12.5 | |

**Note:**
Below the diagonal is the COI sequence; 16S sequence above the diagonal.

All specimens were preserved in the Animal Herbarium of the College of Life Sciences, Xinjiang Agricultural University.

**Type locality and distribution:** Near Kangsu Town (39.80N, 74.73F, altitude: 2,717 m), Wuqia County, Kizilsu Kyrgyz Autonomous Prefecture, Xinjiang Uygur Autonomous Region, China.

**Etymology:** Type specimen locality.

**Diagnosis:** No axillary spots, two nasal scales, upper nasal scale bulging, nostril opening between two nasal scales; three inter-nasal scales, middle one largest, and bulging. Five to seven pairs of orange-red spots along the centre of the dorsal spine and a distinct orange-red spot on the dorsal surface of the caudal base. Dark transverse stripes on the dorsal surface of the tail, white ventral surface of the tail, and black tip.

**Description of holotype (Fig. 6, Table 2):** Adult males (CX09247), SVL = 38.85 mm, HL = 11.43 mm, HW = 11.50 mm, HD = 8.27 mm, RED = 3.61 mm, MB (mouth breadth) = 10.56 mm, NBD = 1.69 mm, LHF (trunk length) = 18.77 mm, TL = 50.12 mm, LFL (length of forelimb) = 23.65 mm, and LHL (length of hindlimb) = 35.30 mm. Bluntly rounded head, with length and breadth not significantly different, and a short and blunt muzzle. One parietal eye with a translucent membrane in the centre surrounded by several irregular scales. Two nasal scales, with the upper nasal scale expanded into a kidney shape, a nostril opening between the two nasal scales, and the nostril opening slanting downwards; three inter-nasal scales, the middle one inflated and bulging and other two smaller and slightly attached to the supra-nasal scales; four scales between the nasal and

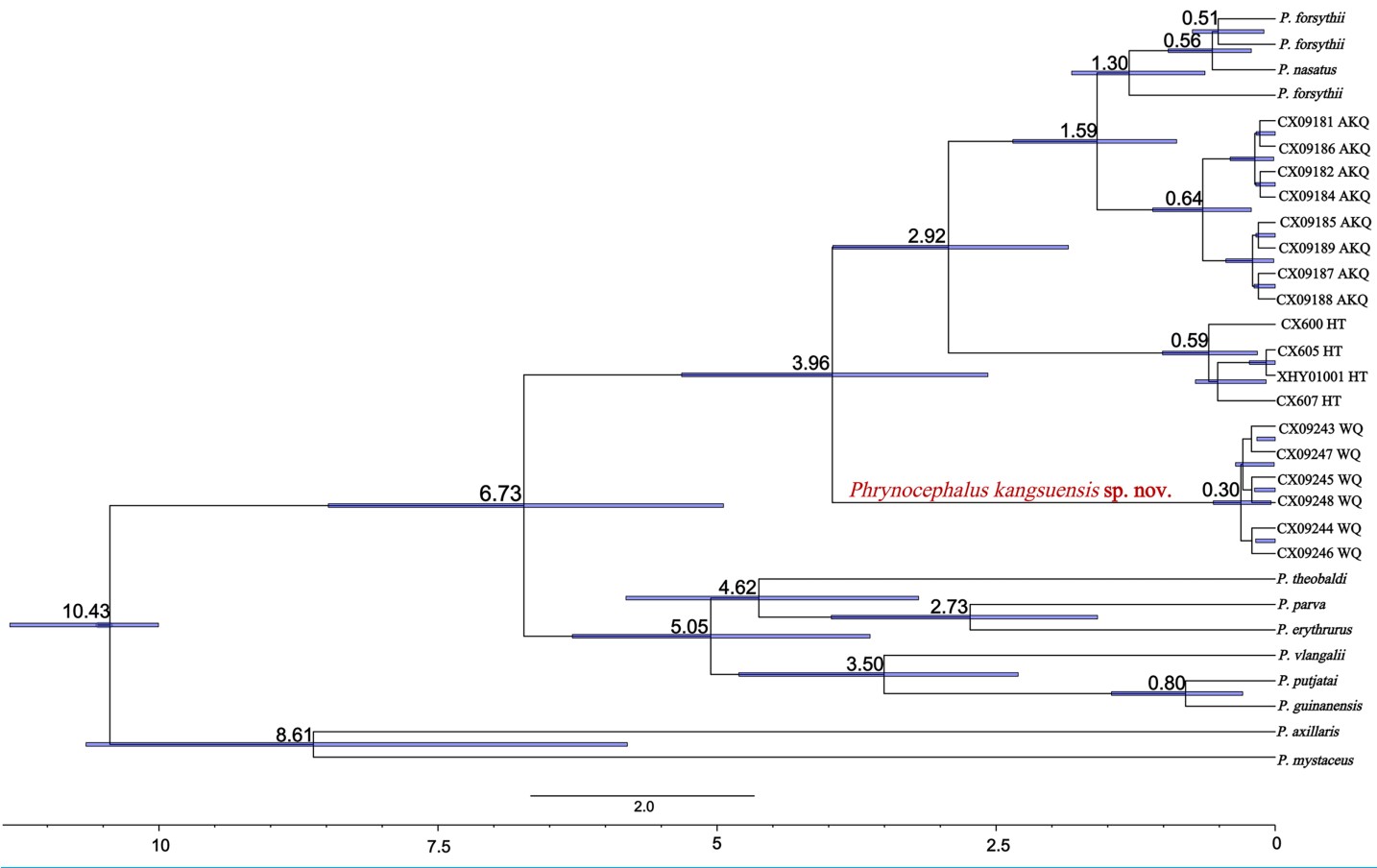

**Figure 5 Divergence time estimates for *Oreosaura*.** Note: Values represent estimated divergence time, and the blue bar represents 95% confidence interval HPD; WQ, Wuqia Group; HT, Hotan Group; AKQ, Akqi Group.

preocular scales and 11 superciliary scales; squarer supralabial shields, with 13 on the left and 12 on the right, and three rows of scales separating the supralabial shields from the lower edge of the eye; 12 infralabial shields, flush with supralabial shields, and 39 chin scales in a longitudinal row in the throat fold.

Axillary is short and dorsal scales are mostly granular, with a few tufts of spiny scales; rhombic abdominal scales, pointed posterior spine corners without ridges, and black centre of abdomen. Short limbs; smooth dorsal scales of limbs; carinate on thigh and shin flanks; rhombic and smooth ventral scales of limbs; and ribbed sub-metacarpal scales. Relatively well-developed and unilateral toe fringe of toe IV of the hind limb. The tail is covered with granular scales, with a few prickly scales; the ventral surface of the basal end of the tail has conical scales and the caudal side is covered with prickly scales.

Grey dorsal surface, densely covered with white dots, with six pairs of orange-red spots along the centre of the dorsal spine from the neck to the base of the tail and clumps of spiny scales on the orange-red spots. No axillary spots; black dots distributed on the dorsal surface of the limbs, particularly on the palms and toes. A distinct orange-red spot on the dorsal surface of the caudal base, nine dark transverse spots on the dorsal surface of the tail

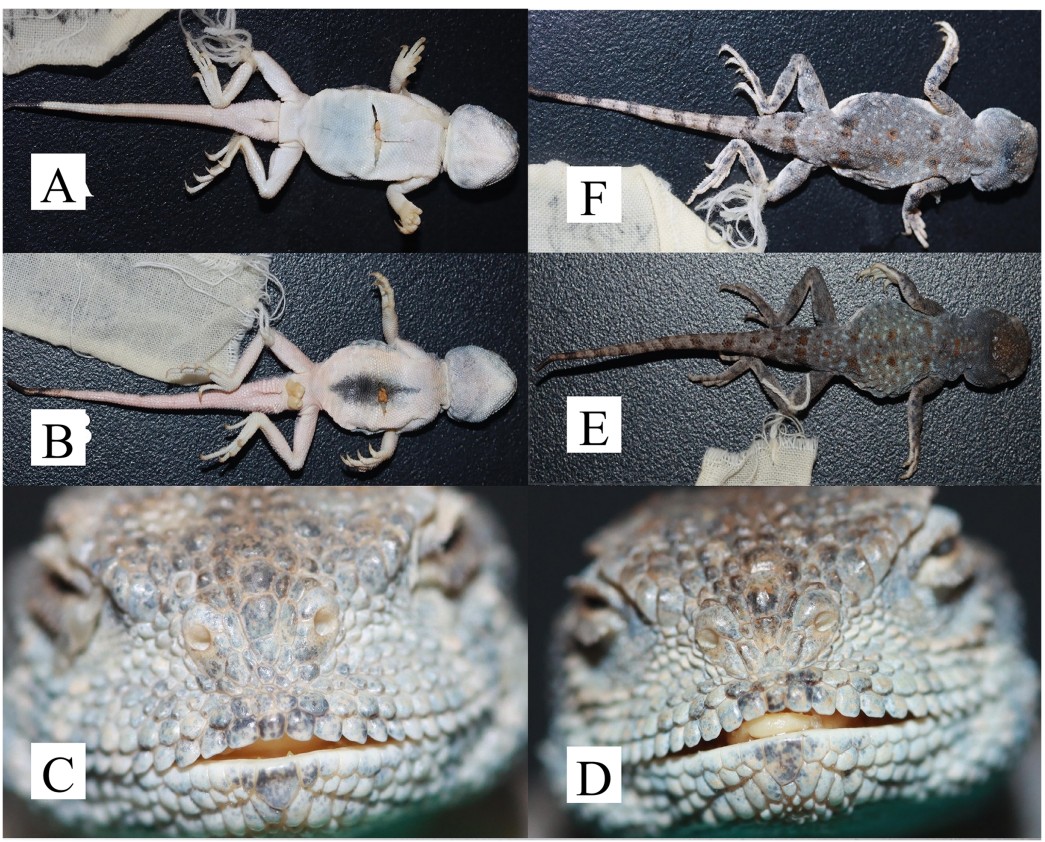

**Figure 6 Pictures of *Phrynocephalus kangsuensis* sp. nov.** (A, C, F) CX09246; (B, D, E) CX09247.

**Table 2 Some measurements (mm) and features of pholidosis of *Phrynocephalus kangsuensis* sp. nov.**

| Numble | sex | 1 | 2 | 3 | 4 | 5 | 6 | 7 | 8 | 9 | 10 | 11 | 12 |
|---|---|---|---|---|---|---|---|---|---|---|---|---|---|
| CX09246 | F | 36.41 | 10.24 | 10.99 | 7.42 | 2.59 | 1.47 | 39.11 | 2 | 3 | 13/14 | 13/13 | 3 |
| CX09245 | M | 39.91 | 12.34 | 11.62 | 7.85 | 3.13 | 1.91 | 49.81 | 2 | 3 | 13/13 | 12/13 | 3 |
| CX09243 | M | 37.65 | 10.91 | 10.38 | 7.25 | 2.53 | 1.79 | 43.73 | 2 | 3 | 13/12 | 12/12 | 3 |
| CX09248 | M | 37.57 | 10.98 | 11.07 | 7.91 | 2.85 | 1.84 | 48.44 | 2 | 3 | 12/14 | 12/13 | 3 |
| CX09244 | M | 38.17 | 11.28 | 10.73 | 7.13 | 2.48 | 1.67 | 48.37 | 2 | 3 | 13/14 | 13/13 | 4 |
| CX09247 | M | 38.85 | 11.43 | 11.50 | 8.27 | 3.61 | 1.69 | 50.12 | 2 | 3 | 13/12 | 12/12 | 3 |

Note:
F, female; M, male; 1, SVL; 2, HL; 3, HW; 4, HD; 5, RED; 6, NBD; 7, TL; 8, NS; 9, IS; 10, supralabial shields (left/right); 11, infralabial shields (left/right); 12, number of scales between lower border of eye and row of supralabial shields.

derived from the end of the tail, ventral surface white, faintly reddish, and the tip of the tail black.

**Variation among Paratypes (Table 2):** SVL = 36.41–39.91 mm; HL = 10.24–12.43 mm; HW = 10.38–11.62 mm; HD = 7.13–7.91 mm; RED = 2.48–3.13 mm; MB = 9.09–10.63 mm; NBD = 1.47–1.91 mm; LHF = 16.85–17.77 mm; TL = 43.73–49.81 mm; LFL = 20.12–22.16 mm; and LHL = 29.72–32.70 mm. There were 5–7 pairs of orange spots

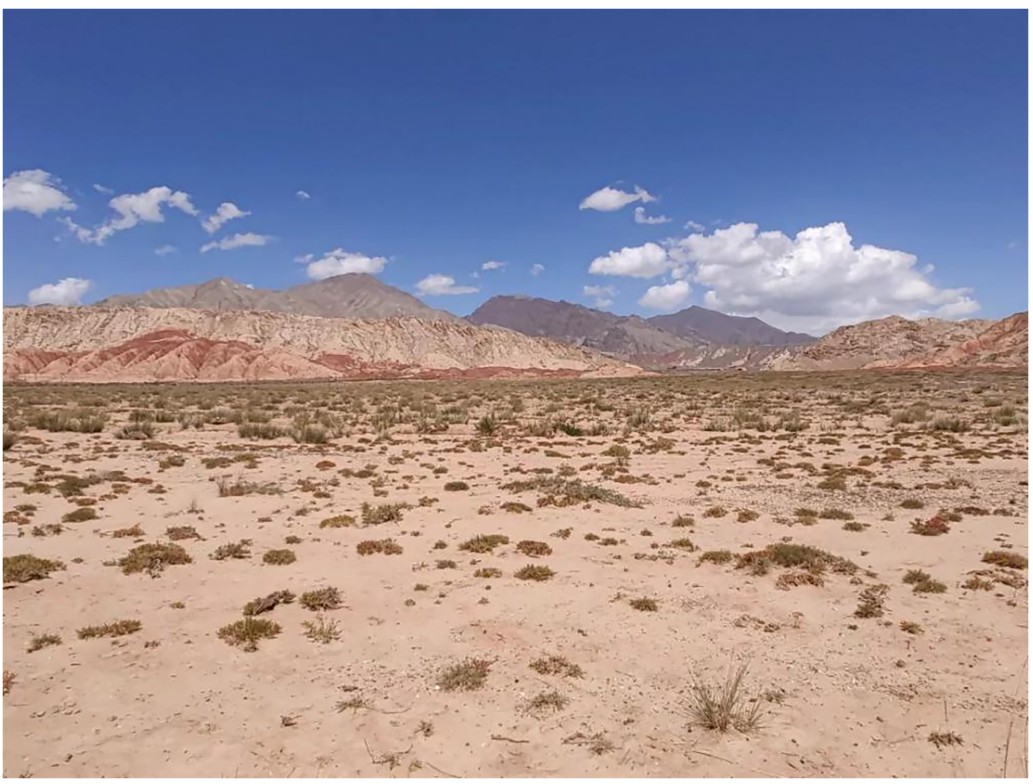

**Figure 7** Type locality of *Phrynocephalus kangsuensis* sp. nov.

on the back. Adult female (CX09246) had a white ventral surface without black spots and 3–5 scales between the nasal and preocular scales, 11–12 superciliary scales, 12–14 supralabial scales, 12–13 infralabial scales, 3–4 scales between the infraocular margin and supralabial scales, and 6–9 dark transverse spots on the dorsal surface of the tail.

**Comparison with other species:** This species exhibits morphological differences compared with those of *P. forsythii* and *P. nasatus*. The relatively widely spaced snout and distinctly inflated and raised inter-nasal scales of *P. kangsuensis* **sp. nov.** are different from those of the other two species. Second, the pattern of spots on the dorsal surface and the transverse stripe on the dorsal surface of the tail are important features for distinguishing the three species. *Phrynocephalus kangsuensis* **sp. nov.** and *P. nasatus* have outwards-facing nostrils with a relatively wide nasal spacing. In addition, *P. nasatus* has 5–6 small inter-nasal scales, while *P. kangsuensis* **sp. nov.** has three inter-nasal scales but the middle one is significantly larger and raised. In contrast, *P. forsythii* has a relatively small nasal spacing, with 3–4 inter-nasal scales of equal size. *Phrynocephalus forsythii* and *P. nasatus* have distinct 4–5 pairs of black dorsal blotches, whereas *P. kangsuensis* **sp. nov.** has 5–7 pairs of orange-red dorsal blotches. The dorsal surface of the tail of *P. kangsuensis* **sp. nov.** has 6–9 dark transverse stripes, while that of *P. nasatus* has distinct 8–9 black transverse stripes and *P. forsythii* has no dark transverse stripes.

**Ecology and Environment (Fig. 7):** This species lives in valleys and is a typical desert animal. The habitat is located in desert flatlands between high mountains. It is a typical arid Gobi environment with a substrate of sandy gravel covered with short grasses.

## DISCUSSION

### Phylogenetic trees

Viviparous species in *Phrynocephalus* are considered to be the subgenus *Oreosaura* (*Barabanov & Ananjeva, 2007*), which includes *Phrynocephalus* spp. distributed at high altitudes. Based on geographic distribution and morphological differences, *Oreosaura* can be divided into three species: *P. forsythii* (*P. forsythii* and *P. nasatus*), *P. vlangalii* (*P. vlangalii*, *P. guinanensis*, *P. putjatia*, and *P. nanschanicus*), and *P. theobaldi* (*P. theobaldi*, *P. erythrurus*, *P. lhasaensis*, and *P. parvus*) (*Solovyeva et al., 2023*).

In the viviparous species group, only *P. forsythii* was distributed on the periphery of the Tarim Basin at lower elevations in southern Xinjiang. As the *P. forsythia* complex contains the only viviparous *Phrynocephalus* species distributed at low altitudes, it has received constant attention from researchers (*Qi et al., 2019*; *Chen, Li & Guo, 2019*; *Qi et al., 2020*, *2021*, *2023*). Based on the phylogenetic tree constructed by *Solovyeva et al. (2023)* for *Phrynocephalus* using COI sequences, we added molecular data from three valleys geographic populations of *P. forsythii* to explore their taxonomic position in the subgenus *Oreosaura*. Our results showed a topology similar to that reported by *Solovyeva et al. (2023)*. Within the *P. forsythii* complex, the three valleys geographic populations (groups WQ, HT, and AKQ) formed three clades with strong support and sister clades to the reference sequences from lower altitudes (Qiemo, Ruoqiang, Kashgar, and Aksu regions). Simultaneously, *P. nasatus*, a lizard collected from Baicheng County, was embedded in *P. forsythia*, and its status needs to be further studied. Second, we added 16S sequences and analyses of spliced sequences from the two datasets as a supplement and the results were consistent with the phylogenetic tree of the COI sequences, with the three valleys populations forming strongly supported clades. Genetic differentiation among several populations in the ~3,000 km ring of the Tarim Basin was extremely low, with significant differences from populations in the closer high-altitude alpine valleys. Geographic populations distributed at valleys were genetically differentiated from populations at plains altitudes.

### Genetic distances

*Solovyeva et al. (2023)* analysed the genetic distances of COI sequences of all species of *Phrynocephalus* and found that the genetic distances between the *P. forsythii* complex and other species in the subgenera were 7.18–8.75% and with *P. axillaris* and *P. mystaceus* were 10.59% and 12.42%, respectively, while the genetic distances between different subgenera exceeding 10%. This was based on the results of *Solovyeva et al. (2023)* and their criteria for genetic differences within species (subgenus >10%, interspecies >5% and intraspecies <3%). Among them, the genetic distance between the Wuqia population (*Phrynocephalus kangsuensis* **sp. nov.**) and other populations within the *P. forsythii* species group reached interspecific differences (>5%), and that between other populations within the *P. forsythii*

complex (Group HT, Group AKQ, and low-altitude populations) was in line with the results of *Qi et al. (2019)* for various geographic *P. forsythii* populations (0.2–5.1%). We calculated and analysed the genetic distances of the 16S rRNA sequences and found that they were lower than those of the COI sequences; however, the results also showed significant genetic differences between the Wuqia population and other populations of the *P. forsythia* complex.

## Species delimitation

The three species delimitations used in this study yielded different results, with ABGD and ASAP delimiting the most similar MOTUs and bPTP delineating additional species. *Solovyeva et al. (2023)* used five methods of species delimitation to define species delimitation in *Phrynocephalus*. They concluded that the ASAP algorithm best reflected the current classification of *Phrynocephalus* and that bPTP grossly overestimated the number of species. Some studies on species delineation recognised that ABGD and ASAP are more conservative and accurate in delineating species, whereas bPTP suffers from over delineation (*Blair & Bryson, 2017*; *Goulpeau et al., 2022*; *Guo & Kong, 2022*; *Ranasinghe et al., 2022*). In the present study, the results of the ABGD and ASAP analyses were consistent with those of *Solovyeva et al. (2023)*. Therefore, the results of our delimitation exhibited a high degree of confidence that the Wuqia and Hotan populations in the *P. forsythii* complex may be hidden species.

## Divergence date estimation

In *Oreosaura*, the *P. forsythii* complex diverged first, followed by the *P. vlangalii* complex and *P. theobaldi* complex, which may be related to the evolutionary relationships of *P. forsythii* and the timing of its geographic occurrence (*Guo & Wang, 2007*; *Jin & Brown, 2013*; *Solovyeva et al., 2018*, *2023*). *Guo & Wang (2007)* suggested that the viviparous species group diverged at 9.12 ± 1.44 Ma; *Jin & Brown (2013)* suggested that the oviparous and viviparous species groups diverged at 9.73 Ma. The results of the divergence time estimation in this study were similar to those of two previous (*Guo & Wang, 2007*; *Jin & Brown, 2013*). *Solovyeva et al. (2018)* suggested that the formation of the subgenus *Oreosaura* was associated with the rapid uplift of the Qinghai-Tibetan Plateau (QTP) and the associated evolution of the East Asian monsoon climate and that the group of viviparous species lived on the QTP from 13.5–10.0 Ma, with differentiation beginning at 3.8 Ma. The timing of the divergence of the viviparous species group was closely related to the uplift of the QTP (*Macey et al., 2018*) and the formation of mountain basins on the QTP at 5.7 Ma may have contributed to significant climate change (*Li et al., 1996*). In addition, the rapid uplift of the Tibetan region at approximately 5 Ma (*Shackleton & Chengfa, 1988*) may have led to the differentiation of *P. theobaldi* complex and *P. vlangalii* complex distributed on the QTP, with the uplift of the Himalayas and the Kunlun Mountains contributing to the formation of species in both species complexes. In their biogeographic study of the *P. theobaldi* complex, *Jin, Liu & Brown, 2017* suggested that the first species divergence of *P. theobaldi* occurred during a possible period of tectonic uplift in the Himalayan region (3.6–7.4 Ma). *Jin, Brown & Liu, 2008* suggested that the

divergence of the *P. vlangalii* complex was caused by three major periods of plateau uplift in the QTP (3.4 Ma, 2.5 Ma, and 1.7 Ma), and the results of the present study support these inferences.

*Solovyeva et al. (2018)* suggested that the progression of aridification, basin reduction, orogeny, and Upper Pleistocene climatic oscillations in the late Miocene may have contributed to the further diversification of *Phrynocephalus*. The divergence of the *P. forsythii* complex is closely related to the uplift of the QTP, Kunlun Mountains, and Tian Shan Mountains. *Guo & Wang (2007)* suggested that the *P. forsythii* complex diverged at 4.35 ± 0.65 Ma, while *Jin & Brown (2013)* suggested that an earlier collapse of the interior of the QTP, which impacted the environment north of the QTP, led to the separation of the *P. forsythii* complex from the other two in the viviparous species group at 5.04 Ma. The present study suggests that the *P. forsythii* complex segregated at 6.73 Ma (95% confidence interval: 4.75–8.76 Ma), which may be related to climate change in the Tarry Basin during the period of ~5–7 Ma (*Sun et al., 2017*).

*Phrynocephalus kangsuensis* **sp. nov.** is found in the northwestern part of the QTP, bordered by the southern foothills of the Tien Shan and the Western Kunlun and Pamir Plateau knots. It inhabits habitats in the Kangsu Valley, which are different from populations living in the lower-altitude plains. The major uplift of the northwest part of the QTP at 4.5 Ma (*Zheng et al., 2000*) and its mutual collision with the Pamir Plateau led to the formation of mountains in the NW part of the QTP, which formed a distinct high and low relief with the Tarim Basin; this may have led to the separation of *P. kangsuensis* **sp. nov.** from other geographic populations.

The Hotan population is located on the northern slopes of the Kunlun Mountains in a high-altitude montane habitat (Karakash Valley). The estimated divergence time for this population is 2.92 Ma (95% HPD: 1.75–4.16 Ma), which may be related to the rapid uplift of the Middle Kunlun area on the northern margin of the QTP since the late Pliocene (4.2–3.9 Ma). The rapid uplift of the Kunlun region led to the vertical geographic isolation of this population from lower-altitude populations, which ultimately led to population differentiation over time.

The Central Kunlun area on the northern margin of the QTP also experienced an uplift in the middle Early Pleistocene (1.66 ± 0.31 Ma) (*Bai et al., 2003*). Second, the uplift of the Tien Shan Mountains during the Early Pleistocene formed the present-day alpine topography (*Sun, Zhu & Bowler, 2004*), which coincides with the timing of species divergence in the AKQ population (Toshkan Valley) and is situated in high-altitude mountainous terrain. The Quaternary climate history of the QTP is characterised by four major ice ages, the most extensive of which was the Naynayxungla Ice Age (0.50–0.72 Ma) (*Zheng, Xu & Shen, 2002*). At the time, many large ice caps, glacial complexes, and grand canyon glaciers formed, which may have facilitated further differentiation of the different alpine valley *P. forsythii* complexes. *Qi et al. (2021)* analysed low-altitude populations in the *P. forsythii* complex and concluded that, in addition to the uplift of the plateau, the rivers and oases of the Tarim Basin and climatic variations, had a marked effect on population differentiation. Since the late Pliocene, Increased aridity led to changes in the vegetation of the Tarim Basin and rainfall and snowmelt on mountaintops have caused

oases to appear in some of the foothill plains, allowing low-altitude populations of *P. forsythii* to form a ring around the Tarim Basin (*Qi et al., 2020*).

Since we did not collect samples from lower altitudes. A limitation of this study is that there was not a large amount of molecular data on *P. forsythii* from low altitudes. Only three of the most representative methods of species delimitation were used. In future studies, more populations of *P. forsythii* from plains could be added and more methods of species delimitation could be used to define the species of the *P. forsythii* complex to allow the specific effects of geological history on *P. forsythii* complex to be studied in further detail.

## CONCLUSIONS

Here, we constructed a COI phylogenetic tree of *Phrynocephalus* based on *Solovyeva et al. (2023)* with additional data. In addition, we constructed a phylogenetic tree of 16S sequences to explore species affinities and divergence events among *Oreosaura*, including within the species complex, as well as between three populations of *P. forsythii* from high-altitude geographic populations.

The results indicate that *Oreosaura* is a strongly supported monophyletic lineage that diverged at 10.43 Ma (95% HPD: [10.00–11.33] Ma), while the *P. forsythii* complex diverged at 6.73 Ma (95% HPD: [4.75–8.76] Ma). The results confirmed that high-altitude valleys promoted the geographic isolation of the *P. forsythii* complex, ultimately leading to species divergence and promoting the formation of new species. The Wuqia population is geographically isolated from the other populations, which leads to genetic differentiation. The molecular results indicated that the Wuqia population is a strongly supported monophyletic lineage with an earlier estimated divergence time than the other populations, at 3.96 Ma (95% HPD: [2.57–5.40] Ma). In addition, this population showed significant differences in morphology.

This study combined molecular and morphological aspects into an integrated taxonomic approach and identified the Wuqia population as a new species, *Phrynocephalus kangsuensis* **sp. nov**.

The results will provide data for phylogenetic studies following the *P. forsythii* complex and data to confirm that valleys promote the formation of species differentiation.

## ACKNOWLEDGEMENTS

We extend our gratitude to our colleagues in the project team for their valuable technical assistance. We would like to thank Dong-Liu Meng for helping to record the data during the morphometric measurements, and Wei-Zhen Gao for providing guidance on photography. We would like to thank Editage for English language editing.

### Funding

This work was supported by the Third Xinjiang Scientific Expedition Program [Grant No.2022xjkk1200] and the National Natural Science Foundation of China

[Nos.31660613]. The funders had no role in study design, data collection and analysis, decision to publish, or preparation of the manuscript.

## Grant Disclosures
The following grant information was disclosed by the authors:
Third Xinjiang Scientific Expedition Program: 2022xjkk1200.
National Natural Science Foundation of China: 31660613.

## Competing Interests
The authors declare that they have no competing interests.

## Author Contributions
- Qianru Liang conceived and designed the experiments, performed the experiments, analyzed the data, prepared figures and/or tables, authored or reviewed drafts of the article, and approved the final draft.
- Lei Shi conceived and designed the experiments, authored or reviewed drafts of the article, and approved the final draft.

## Animal Ethics
The following information was supplied relating to ethical approvals (*i.e.*, approving body and any reference numbers):

All experimental procedures involving animals were approved (animal protocol number: 2023013) by the Animal Welfare and Ethics Committee of Xinjiang Agricultural University, Urumqi, Xinjiang, China.

## DNA Deposition
The following information was supplied regarding the deposition of DNA sequences:

The sequences are available in the Supplemental File and at NCBI: OR884245-OR884250, OR878654-OR878659, OR875280-OR875291, OR875272-OR875275, OR875264-OR875271.

## New Species Registration
The following information was supplied regarding the registration of a newly described species:

Publication LSID: urn:lsid:zoobank.org:pub:812FA766-6BC6-4D39-85B9-8E2C58E06098

*Phrynocephalus kangsuensis* **sp. nov.** *phrynocephalus kangsuensis* sp. nov. Qianru & Shi, LSID: urn:lsid:zoobank.org:act:53EF51A0-DEEA-4E6A-B62F-AD61AB3868E9

## Supplemental Information
Supplemental information for this article can be found online at http://dx.doi.org/10.7717/peerj.17175#supplemental-information.

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
