# Peer review of "Species divergence in valleys: the phylogeny of Phrynocephalus forsythii complex and description of a new species"

_PeerJ, doi:10.7717/peerj.17175_

## Round 0.1 · original submission · Major Revisions

Please, include the answers to all reviewers' comments. Reviewer 1 raised several issues, which have to be carefully addressed.

**Language Note:** The review process has identified that the English language must be improved. PeerJ can provide language editing services - please contact us at [email protected] for pricing (be sure to provide your manuscript number and title). Alternatively, you should make your own arrangements to improve the language quality and provide details in your response letter. – PeerJ Staff

Reviewer 1 ·

Basic reporting

This research presented some new data on the morphology and genetics of P. forsythii. The authors try to descript a new species of Phrynocephalus kangsuensis sp. nov. from Wuqia, Xinjiang. While there is some new, there are many clear weak points which need to be addressed and changed before its acceptance in any journal. I raised the followings major concerns:

1. Solovyeva et al. (2023) is to review the hidden diversity of Phrynocephalus using short mtDNA fragments wich clear weakness, the data they published could not presented clear evidence on the species delimitation on some specific lineages, i.e., the Chinese lineages. However, the authors of this work only continued to use two similar short mtDNA fragments (COI and 16sRNA), and conducted similar species delimitation analyses with any nuclear DNA information, and did not compare morphological difference. This clearly impede the robust statistics of general species delimitation. BPP or BP values is not a nice value used for species justification without other delimitation analyses. The authors should pay enough caution to raise a new species name without clear and strong supports.
2. The authors need pay attention on why COI and 16sRNA presented different topology of P. forsythii lineages.
3. The authors need demonstrat the reasons on how to divide the populations into higher or lower populations. Really, the detail geographic locality information need to be addressed in the attached files, such as the altitude, longitude and latitude of the sampling localities. Administrative region description is not clear.
4. Why only show the pictures of male? How about females? It is very important to show the detail geographic latitude and longitude information of type locality, and show morphology of both sexes, and try to show the detail morphology difference to support its independent species status.
5. There were some incorrect positon for some species in there topology, i.e., P. vlangalii, it is the most recent emerged taxa in all viviparous Phrynocephalus. While the focus of their manuscript is rely on topology of P. forsythii, and the current data is not enough to conduct robust phylogenetic analyses with more Phrynocephalus, I suggest to delete other species except P. forsythii and just to keep one oviparous outgroup species to conduct their phylogenetic analyses.
6. Overall, I tend to see a new version on the research of morphological and genetic divergence and dating of P. forsythii, with some changing on the title.

Experimental design

Reasonable wish some weak point raised above.

Validity of the findings

Generally, not very interesting, but it is worthy to publish if it could be justified well.

Additional comments

none

·

Basic reporting

Additional English proofreading is needed, the meaning is not clear everywhere in the text.
Field background and sufficient introduction into the area were provided.
All necessary figures, tables and raw data were provided. Although some parts of Results section should be moved to Materials and Methods section.
Results are relevant with the hypotheses.

Experimental design

no comment

Validity of the findings

no comment

Additional comments

Methods were used properly, the research provided new and interesting results, including the description of a new species. But the text needs some improvements, including additional English proofreading.

29 - cytochrome coxidase → cytochrome c oxidase
36-38 - “The results will provide data for phylogenetic studies following the P. forsythii complex, as well as data to confirm that valley do indeed promote the formation of species differentiation.” - the meaning is not clear. Do you mean that exact valley or the role of valleys Phrynocephalus species formation?
Materials and Methods section – please, unify the tenses you use. For example: “All
specimens are deposited…”, but “…sequences available for other species of the subgenus Oreosaura were collected...”
96 - “The liver tissue in the samples is preserved with 95% ethanol and for DNA extrcation” → Did you use only liver tissue? May be than it will be better to rephrase.
DNA extrcation → DNA extraction
100 - “We also adhere to the ARROW guidelines” – expand, what means the abbreviation ARROW
113-114 – not genes were stored, amplicons. Please, improve

115-116 – what do you mean by sequence splicing? Concatenation?
136 – you probably missed values for BPP, there are > 0.99
153 – dPTP → bPTP (check it further in text)
What are the outgroups you used in the phylogenetic analyses? Please, mention them in the M&M section

Results section
184 – “base sequences” – do you mean single genes analyses?
188 – please, unify tenses
188 – “was differentiated” - > differentiated
201-202 – “Genetic distances (p-distance) were calculated using MEGA 7.0 for each lineage in the subgenus Oreosaura” – no need to specify here the method. It was mentioned in M&M section.
227 – “...to diverge in the P. forsythii complex...” → within the complex P. forsythii
236 – “...mating specimen consisted of…” - please, clarify, what is a mating specimen
296-297 – May be some word is missing in the phrase (here: “and is typical of an arid Gobi habitat…”)?

Through the text: check the “P. forsythii”, somewhere it is “P. forsythia”.

Figures of phylogenetic trees – please mark somehow branches, which have the highest support values from both analysis.

---

## Round 0.2 · Minor Revisions

Please, address the concerns raised by Reviewer 1 who raised several methodological issues.

Reviewer 1 ·

Basic reporting

The revised version improved a little, unfortunately the revision can not satisfy what my major concerns. Obviously the authors did not provide a robust phylogeny of the genera with their few mitochondrial markers, and they presented conflicted topologies on the main lineages of the complex. The latter clear weak point is that they still can not solve the conflicted monophyletic topologies between 16sRNA and COI/16sRNA spliced markers. In Fig. 3, the new pupative species of WQ population firstly diverged for 16sRNA, but in the spliced topology of other figures, the lineage diverged most recently. The data can not be published as the current version due to the clear chaoes of topologies.
Moreover, the author presented some morphological data of the new proposed species, but still did not compare the morphological divergence among the main lineages of P. forsythii complex, which limited their conclusion of clear morphological divergence of the species. I do suggest the authors to conduct morpholgocial comparison between the proposed new species and other lineages.
The 3% genetic divergences should be in very careful to be used to raised a new spcies without clear morphological divergence in a monophyletic topology. I see the authors used some species delimitation analyses, it is just statistica and might always present new species due to the large mtDNA divergence. The authors need pay attention to the lack of morphological comparision, the potential locus sampling errors between markers, and the clear weak of lacking of nDNA data to raise a new species.

Experimental design

Dating calibration on divergence and the parameter setting in Beast need to be explained well.
It seems that there were lack of clear genetic divergence among individuals from the same population, the haplotypes from spliced data and their corresponding phylogenetic analyses should be more preferred.

Validity of the findings

Lack of robust evidence to their conclusion.

Additional comments

none

·

Basic reporting

no comment

Experimental design

no comment

Validity of the findings

no comment

Additional comments

Dear Authors and Editor,

I'm completely satisfied with the changes, manuscript is ready for publication.

Yours sincerely,
Evgeniya

---

## Round 0.3 · accepted · Accept

Thank you for addressing all the reviewers' comments, especially those regarding methodology. I am happy with the current version and I think the manuscript is ready for publication.